# Quorum sensing as a mechanism to harness the wisdom of the crowds

Stefany Moreno-Gámez[1,2] ✉, Michael E. Hochberg[3,4] & G. S. van Doorn[1]

Bacteria release and sense small molecules called autoinducers in a process known as quorum sensing. The prevailing interpretation of quorum sensing is that by sensing autoinducer concentrations, bacteria estimate population density to regulate the expression of functions that are only beneficial when carried out by a sufficiently large number of cells. However, a major challenge to this interpretation is that the concentration of autoinducers strongly depends on the environment, often rendering autoinducer-based estimates of cell density unreliable. Here we propose an alternative interpretation of quorum sensing, where bacteria, by releasing and sensing autoinducers, harness social interactions to sense the environment as a collective. Using a computational model we show that this functionality can explain the evolution of quorum sensing and arises from individuals improving their estimation accuracy by pooling many imperfect estimates – analogous to the 'wisdom of the crowds' in decision theory. Importantly, our model reconciles the observed dependence of quorum sensing on both population density and the environment and explains why several quorum sensing systems regulate the production of private goods.

Quorum sensing is a process whereby bacteria synthesize small molecules known as autoinducers that are either passively or actively released into the extracellular space. These molecules accumulate extracellularly and their concentration is sensed by bacteria via specialized receptors. Upon reaching a threshold concentration, autoinducers trigger cascades of signal transduction which are well described in many bacterial species and regulate processes such as biofilm formation, virulence, competence and sporulation[1–3].

Despite the detailed understanding of the molecular mechanisms underlying various quorum sensing systems, the adaptive value and evolutionary origin of quorum sensing are less understood[4,5]. The prevailing functional interpretation of quorum sensing states that bacteria engage in releasing and sensing autoinducers to monitor population density. This would ensure that individuals express quorum sensing-regulated traits only when there is a sufficiently high number of other individuals also expressing them (hence the term 'quorum')[2,3]. However, this explanation is based on two premises that

have been challenged in light of accumulating evidence on the diversity and complexity of quorum sensing systems.

The first premise is that the benefit that an individual gains from expressing a quorum sensing-regulated trait increases with population density. There is evidence of this idea in systems where quorum sensing controls the production of 'public goods' (e.g. extracellular proteases). In this context, secreting costly molecules is more efficient if other cells engage in the same behavior and thus the benefit of upregulating these traits increases with the number of cells[6]. However, in other systems quorum sensing primarily regulates the expression of 'private' functions such as competence or persistence that are not shared with other individuals[3,7–9]. Since private functions -unlike public goods- can be beneficial for a cell regardless of the number of neighboring cells expressing them, it is less clear why bacteria should regulate these functions by monitoring population density.

The second premise is that bacteria can reliably estimate population density by sensing local autoinducer concentrations.

[1]Groningen Institute for Evolutionary Life Sciences, University of Groningen, P.O. Box 11103, 9700 CC Groningen, The Netherlands. [2]Department of Civil and Environmental Engineering, Massachusetts Institute of Technology, Cambridge, MA 02139, USA. [3]ISEM, Université de Montpellier, CNRS, IRD, EPHE, 34095 Montpellier, France. [4]Santa Fe Institute, Santa Fe, NM 87501, USA. ✉e-mail: stefany@mit.edu

This assumption has been notably challenged by studies in different quorum sensing systems demonstrating that the relationship between cell density and the concentration of autoinducers can be contingent on environmental conditions. The best-known environmental factor mediating this relationship is the diffusivity of the extracellular environment. For instance, at sufficiently low diffusivity, autoinducer concentrations can result in the quorum for quorum sensing induction to be a single cell[10]. These and other observations led to the 'diffusion sensing' hypothesis, which states that bacteria release autoinducers to test environmental diffusivity and regulate the secretion of costly molecules into the extracellular environment[11]. This hypothesis was later reformulated as 'efficiency sensing' to acknowledge that bacteria cannot disentangle local cell density from environmental diffusivity using the concentration of autoinducers but instead rely on both factors to determine the efficiency of producing costly diffusible molecules. Nevertheless, emphasizing diffusion as the main functional driver of quorum sensing likely underestimates the complexity of quorum sensing regulation given that many other factors such as pH, oxygen and antibiotic stress can influence quorum sensing as well[12–14]. An alternative, more integrative perspective acknowledges that several biotic and abiotic factors regulate quorum sensing systems and that responding to a combination of these factors rather than to a single one better explains the functional role of quorum sensing for bacteria in nature[4,14,15].

If quorum sensing is indeed an adaptive mechanism to respond to a combination of abiotic and biotic environmental factors, this raises the question of why bacteria would evolve to employ collective sensing of environmental information over direct individual sensing. Here we propose that bacteria benefit from regulating gene expression through quorum sensing because cell-to-cell communication allows individuals to collectively determine the state of the environment by pooling information at spatially relevant scales. This in turn enables them to make more reliable decisions about when to upregulate the expression of quorum sensing-controlled traits. According to this hypothesis, cells sense their environment using various mechanisms and encode this information in the rate of autoinducer production - an assumption supported by observations from multiple quorum sensing systems[13,14,16–20]. Then, by secreting autoinducers and monitoring their extracellular concentrations, cells can share private estimates of environmental conditions and gain access to a 'pooled' estimate of the environment – analogous to the "wisdom of crowds" in decision theory, whereby noise in individual estimates of the environment promotes the use of group consensus[21,22]. This hypothesis is not exclusive with the prevailing paradigm that bacteria use quorum sensing to coordinate gene expression at high cell densities. Nevertheless, we show here that the benefits derived from collective sensing are sufficient to explain the evolution of quorum sensing.

## Results

### Model

We study the evolution of quorum sensing in fluctuating environments, where the estimates of environmental conditions by individual bacteria are noisy. Our model assumes the simplest possible internal network of feedback regulation (Fig. 1a), whereby a gene product $A$ promotes its own transcription (Methods). We parametrize this simple positive feedback network such that there are two possible stable states: an 'OFF' state where $A$ is expressed at a low basal level, and an 'ON' state where $A$ is expressed at a higher level than in the OFF state (Fig. S1 and Methods). This is an approximation of systems where bacteria use quorum sensing to modulate all-or-nothing programs of gene regulation that control decisions such as sporulating or becoming competent or virulent[23–26]. We assume that bacteria can exchange $A$ with the extracellular environment by passive diffusion through the cellular membrane, which is how quorum sensing works in many Gram-negative bacteria that do not have dedicated transporters for

quorum sensing signals[27]. Hence, $A$ acts both as a product of the gene regulatory network and as a quorum sensing signal.

In order to study how communication evolves in our model, we let bacteria evolve a parameter $c$ that sets the rate of passive diffusion of $A$ through the cell membrane. This parameter determines the membrane permeability to $A$ and thus the degree of communication between cells (e.g. when $c = 0$, a cell does not share or receive any information from other cells). In nature, membrane permeability depends in part on the biochemical properties of autoinducers and can change because of variations in autoinducer length or molecular structure, as well as by the evolution of active mechanisms for autoinducer secretion or transport (e.g. carrier proteins)[28–30].

We simulate a population of bacteria inhabiting a two-dimensional grid over which $A$ diffuses with diffusion rate constant $D$. Bacteria evolve through a series of environmental cycles that fluctuate randomly between two equally probable alternative states, $E_{OFF}$ and $E_{ON}$ (Fig. 1b). In each environmental state there is an optimal level of $A$ expression for all individuals in the grid: while $E_{OFF}$ favors bacteria that do not express $A$, $E_{ON}$ favors bacteria with high levels of expression. For instance, $E_{ON}$ could correspond to an environment where $A$ activates an adaptive program to cope with stress (e.g., competence). Activating such a program would not be useful in the absence of the stressor ($E_{OFF}$) and thus bacteria would benefit from switching off the production of $A$ in this context (see Table 1 for examples of $E_{ON}$ environments in several quorum sensing systems). At the end of each environmental cycle the performance of an individual is calculated as the absolute difference between the value of $A$ and the current optimal expression level, either $A_{ON}$ for $E_{ON}$ or $A_{OFF}$ for $E_{OFF}$, averaged over the cycle duration. This value is then used to compute individual fitness using a sigmoidal function such that cells are penalized for errors in determining whether the environment is in the ON or OFF state, but not for small numerical deviations from the optimal value of $A$ when $c = 0$ (see Fig. S2 and Methods). Importantly, we assume that matching the state of the environment is the only factor that determines the fitness of a cell. In nature, bacteria might benefit from estimating other quantities from the concentration of autoinducers in addition to the state of the environment (e.g. population density), but we purposely left these aside to focus on the role of collective sensing on driving the evolution of quorum sensing.

Reproduction occurs at the end of each environmental cycle and cells are selected to reproduce with a probability proportional to their fitness. Reproducing cells are sampled with replacement, and offspring are collected until their total number is sufficient to replace the parental generation and fully repopulate the grid. As a consequence, population size remains constant, generations are non-overlapping and individuals can have multiple descendants in the next generation. Upon cell reproduction, $c$ mutates with probability $\mu$, resulting in $c$ increasing or decreasing with equal probability by a fixed step size $\delta$ (subject to the constraint that $c \geq 0$). Finally, daughter cells are placed in the nearest location available to the position of their parent.

Exchanging $A$ with the extracellular environment provides cells with information on the initial extracellular concentration of $A$, which could potentially be beneficial if this concentration is informative of the current environmental state. To prevent such benefits (which do not result from cell-cell communication) from biasing the outcome towards the evolution of high $c$, we implement initial conditions for the extracellular concentration of $A$ that are uninformative to cells. In particular, we assume that, for each generation of cells, the initial concentration of $A$ is sampled independently for each grid cell from a uniform distribution in the interval $[0, A_{OFF} + A_{ON}]$.

Finally, a key assumption of the model is that bacteria can differ in their individual estimates of the environment despite encountering the same environmental regime and having the same internal gene regulation network. Such phenotypic heterogeneity has been documented in several quorum sensing systems (e.g., bioluminescence in

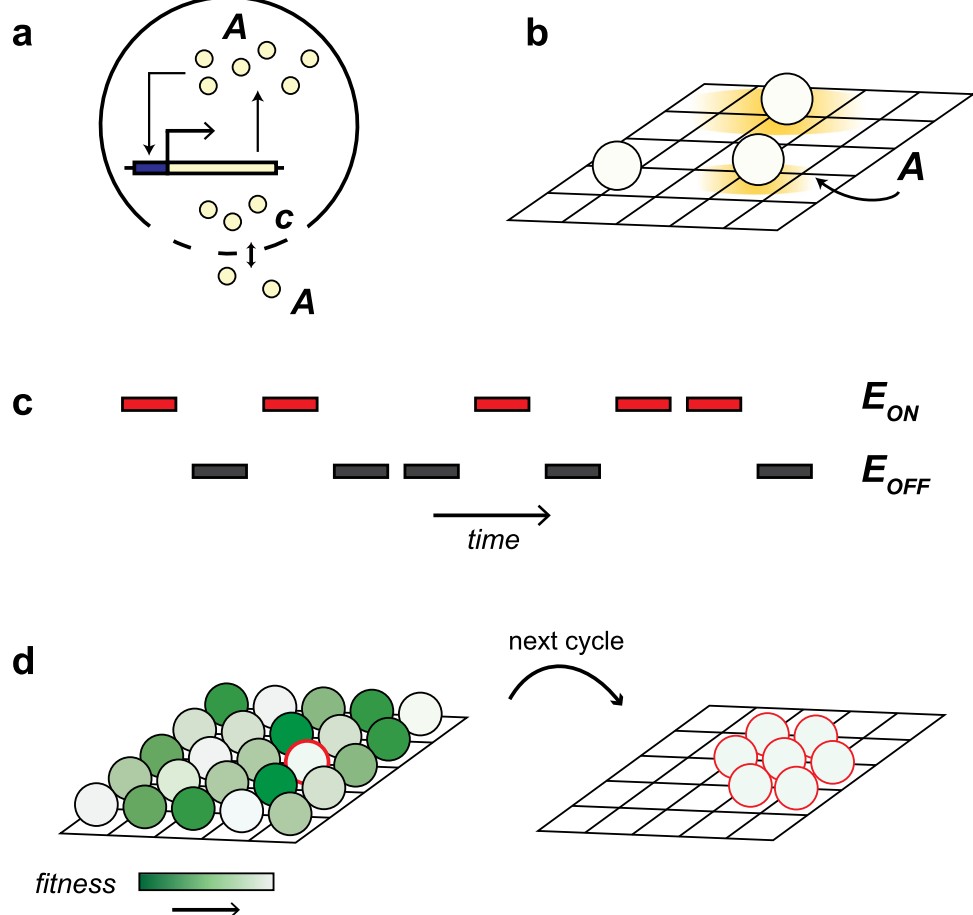

**Fig. 1 | Model structure. a** Internally, every cell has the simplest gene network of positive feedback regulation, where $A$ promotes its own transcription. This network is parameterized as a bistable system with two stable equilibria separated by an unstable equilibrium in the midpoint between both (Fig. S1). Bacteria can also exchange $A$ with the extracellular environment by passive diffusion through the membrane at a rate proportional to the evolvable parameter $c$. **b** At each timestep, the intracellular concentration of $A$ is updated for every cell, and the extracellular concentration of $A$ is updated according to a diffusion process with diffusion constant $D$ over the 2-D grid (bacteria occupy the whole 2-D grid, but only 3 cells are shown for illustration). The sizes of the yellow halo illustrate different scenarios: (bottom) a cell with $c = 0$ that does not exchange $A$ with the extracellular environment; (center) a cell that either has a low value of $c$ or lives in an environment where diffusivity $D$ is low; (top) a cell with a high value of $c$, or that lives in an environment with high diffusivity. **c** The environment experienced by each cell on the grid fluctuates randomly between two states, $E_{ON}$ and $E_{OFF}$. In generations when the environment is in the $E_{ON}$ state, bacteria maximize their fitness by expressing $A$ at a high level, whereas in the $E_{OFF}$ state, fitness is maximal when $A$ is produced at a low basal level. The fitness of every cell is calculated at the end of every generation as the difference between its level of expression of $A$ and the optimal level of expression given the environmental state, averaged over the entire generation. **d** The grid is repopulated such that every individual has the chance of reproducing with a probability proportional to its fitness and its descendants are placed at or in adjacent locations to its position in the grid. This is illustrated for a single high fitness parent and its offspring (fitness increases from green to white). The full grid is repopulated every generation but for illustration only the offspring of one cell is shown.

*Vibrio*, competence in *Bacillus* and virulence in *Listeria*) where actively quorum-sensing isogenic populations contain subpopulations of cells in an OFF state[31,32]. The origin of these phenotypic differences has been partially attributed to stochastic events at the level of expression of quorum-sensing-related molecules, in particular of autoinducers, response regulators, and proteins involved in the cascades of quorum sensing regulation[32–36]. In our model, this cell-to-cell variation is captured by assuming that at the start of an environmental cycle each bacterium makes an individual estimate of the state of the environment that is reflected in its internal $A$ concentration. We implement this by letting bacteria sample their internal $A$ concentration from a (truncated) normal distribution whose mean is the optimal level of $A$ expression in the current environment (either $A_{ON}$ or $A_{OFF}$). Sampling from a distribution reflects the assumption that bacterial estimates of the current environment are noisy (to an extent quantified by the variance in the distribution), due to some combination of environmental unpredictability and intrinsic estimation errors.

## The evolution of communication as a collective sensing strategy

Starting from a population where cells do not share information ($c = 0$ initially), we find that $c$ increases over time and thus communication readily evolves in the population (Fig. 2). Interestingly, although $c$ initially increases slowly, there are successive sweeps that lead to a rapid transition towards higher values of $c$, followed again by a slower increase. This occurs because (i) communication becomes beneficial only after a minimum number of neighboring cells are exchanging information (Fig. S3) and (ii) once most cells are coupled to each other these benefits increase marginally with the mean value of $c$ in the population (Fig. S4). Specific features of these dynamics such as how the advantage of collective sensing depends on the number of communicating cells or the speed at which selective sweeps occur in the population, depend on the shape of the fitness function and the size of the mutational step (Figs. S5 and S6). However, across different parameter values there is a consistent pattern of positive frequency-dependent selection on communication and its eventual stabilization.

We find that a series of conditions favor the evolution of quorum sensing due to its collective sensing functionality. The first two are related to model assumptions justified previously. First, collective sensing is beneficial only if, on average, cells make an individual estimate sufficiently close to the current state of the environment (Fig. S7a). Thus, our model is consistent with a general principle of decision theory known as the Condorcet Jury Theorem. This theorem establishes that for a group of individuals using a majority-rule for decision making, the chance of making the right choice increases with the number of voters only if individuals make the correct choice more often than the incorrect one[37]. Second, provided that on average individual estimates of the environment are correct, increased noise in the individual estimates of the environment facilitate the evolution of collective sensing (Fig. S7b). By contrast, in the extreme scenario where cells could determine the exact state of the environment on their own, there would be no benefit of cell-to-cell communication as a way to improve individual estimates of environmental conditions.

In addition to the previous conditions, we find that the evolution of collective sensing is facilitated by two features of bacterial interactions. First, in the absence of motility, the offspring of a bacterial cell is often located nearby in space. Our model shows that such spatial placement of offspring accelerates the evolution of collective sensing relative to random placement (Fig. 3a, c), a result that is consistent with the considerable literature on the importance of spatial structure in the evolution of collective behavior[38–40]. Collective sensing evolves in our model, because the benefits of cell-to-cell communication are accrued locally through successive environmental generations. Dispersal frustrates this evolution, both in the source habitat of the mother cell and in the target areas to which daughter cells disperse.

Second, similar to dispersion of offspring relative to the parent cell, environmental diffusivity also influences the evolution of collective sensing. However, unlike the monotonic negative effect of cell dispersal, extreme low or high diffusivity hinders the evolution of cell-to-cell communication (Fig. 3 and Fig. S8). On the one hand, in the hypothetical scenario where there is no environmental diffusivity there would be no exchange of information between cells, which in turn would preclude the evolution of high c (Fig. S8). On the other hand, highly diffusive environments tend to couple communicating bacterial cells with many non-communicators, diminishing the benefit of local assortment and slowing the evolution of communication (Fig. 3b, c). Likewise, we find that collective sensing is facilitated if bacteria are in moderately confined environments (Fig. S9). Otherwise, if the extracellular volume is sufficiently large such that bacterial secretion of autoinducers has little impact on extracellular autoinducer concentration, then cell-cell communication through autoinducer secretion is not effective anymore.

Taken together, these results indicate that the evolution of collective sensing is favored when bacterial cells interact locally.

## Table 1 | Examples of $E_{ON}$ environments in different QS systems

| QS-regulated phenotype | $E_{ON}$ |
|---|---|
| Competence | Antibiotic stress, DNA damage[14,17,58,59] |
| Sporulation | Nutrient starvation, DNA damage[7,60,61] |
| Persistence | Oxidative and antibiotic stress, high temperature[62–64] |
| Metabolic rewiring | Nutrient starvation[65–68] |

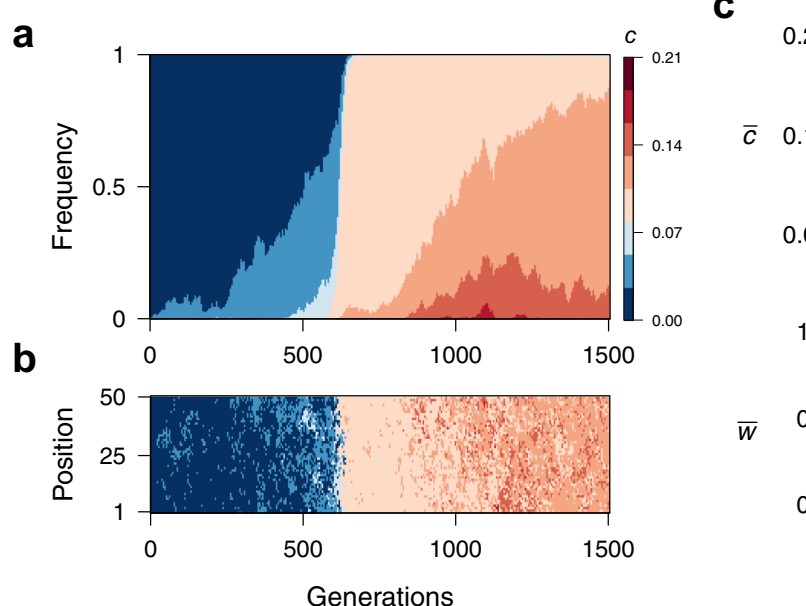

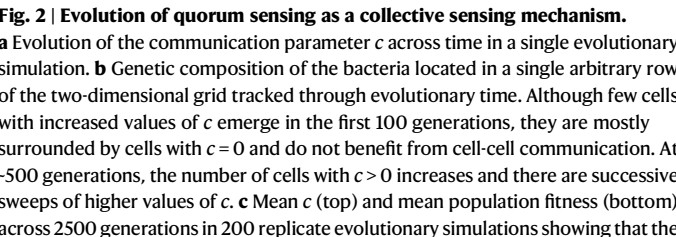

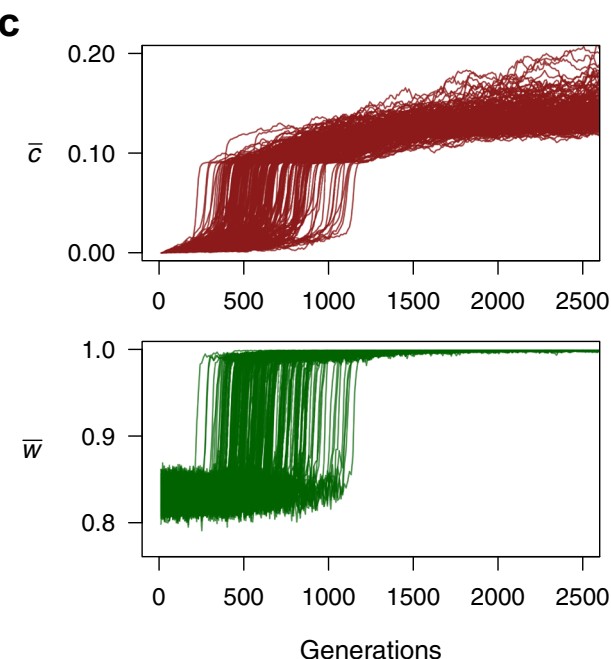

**Fig. 2 | Evolution of quorum sensing as a collective sensing mechanism. a** Evolution of the communication parameter c across time in a single evolutionary simulation. **b** Genetic composition of the bacteria located in a single arbitrary row of the two-dimensional grid tracked through evolutionary time. Although few cells with increased values of c emerge in the first 100 generations, they are mostly surrounded by cells with c = 0 and do not benefit from cell-cell communication. At ~500 generations, the number of cells with c > 0 increases and there are successive sweeps of higher values of c. **c** Mean c (top) and mean population fitness (bottom) across 2500 generations in 200 replicate evolutionary simulations showing that the rapid spread of communication through the population is associated with a rapid increase in the benefit of collective sensing arising once there is a minimum degree of communication in the population. When the mean c exceeds 0.1, most cells are communicating to the extent that they can correctly determine the state of the environment and mean fitness approaches 1.0. Thereafter, and due to the sigmoidal shape of the fitness function, evolving higher c has a marginal effect on fitness. The gene regulatory network is parameterized as shown in Fig. S1 and the rest of parameters are $E_{OFF} = 10$, $E_{ON} = 80$, $\sigma_{OFF} = 25$, $\sigma_{ON} = 50$, $D = 0.5$, $\mu = 0.001$, $\partial = 0.03$, $s = 0.8$ and $x = 20$.

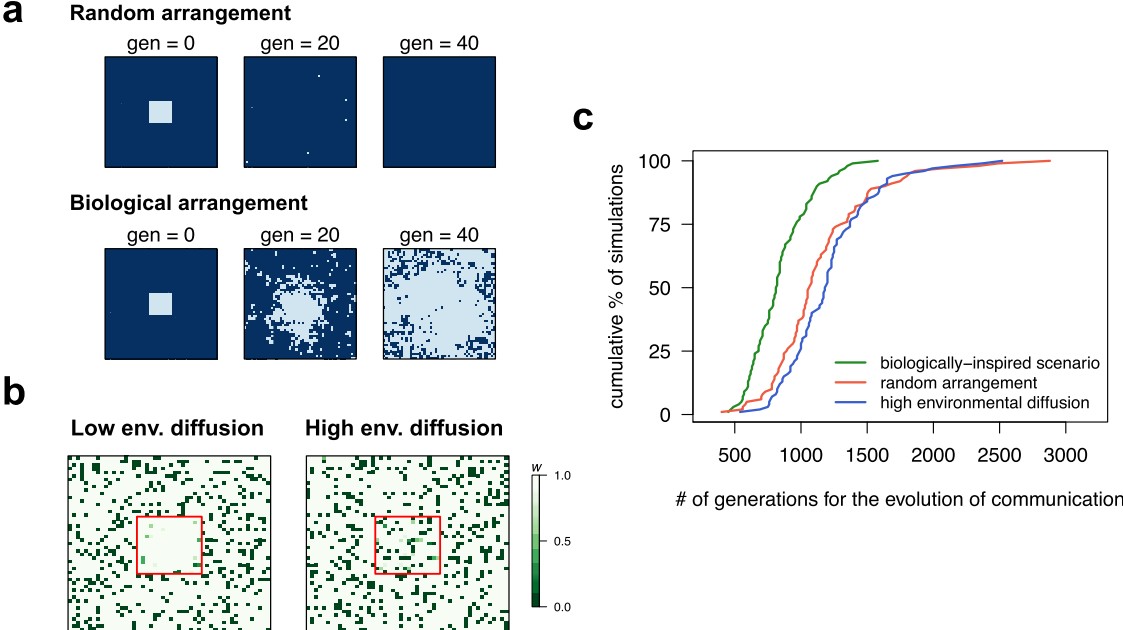

**Fig. 3 | Quorum sensing, local interactions and the role of environmental diffusion. a** Genetic composition of two populations (shown in the two-dimensional 50 × 50 grid) that start with a subpopulation of communicators (light blue, $c = 0.1$) surrounded by non-communicators (dark blue, $c = 0$) across 40 generations of selection. In the top populations, the offspring of a cell is placed randomly on the grid, whereas in the bottom populations, offspring occupy a position close to their mother cell. Random placement of offspring leads to the extinction of communicating cells because bacteria with high $c$ only benefit from collective sensing if there are other communicators nearby. **b** Individual fitness values in two populations of non-communicators ($c = 0$) that contain a subpopulation of communicators ($c = 0.1$, shown by the red square). When environmental diffusion is low ($D = 0.5$), the subpopulation of communicators benefits from collective sensing, whereas at high environmental diffusion ($D = 50$), communicating cells are coupled with non-communicators and any fitness benefit of collective sensing disappears.

Fitness values are calculated after one generation in an $E_{ON}$ environment. A similar pattern is observed in an $E_{OFF}$ environment. **c** Cumulative distribution of the time to evolution of communication in three scenarios: (green) biologically-inspired scenario presented in Fig. 2, where the offspring of a cell remain nearby and bacteria interact locally; (red) same scenario as Fig. 2 except the offspring of a cell are randomly placed over the spatial grid after reproduction; (blue) same scenario as Fig. 2 except the rate $D$ of diffusion in the extracellular environment is high so bacteria have a long interaction range. 100 simulations are shown per condition and we assume that communication evolves when the mean $c$ exceeds 0.1 (Fig. 2c). For all panels, unless indicated otherwise, parameters are as in Fig. 2. Note that the evolution of collective sensing is further hindered if, in addition to high environmental diffusion or random placement of the offspring, the mutational stepsize is large; in either case, we failed to observe the evolution of collective sensing in 7000 generations for 50 replicate simulations with $\partial = 0.1$.

## Collective sensing in spatially structured environments

The patterns identified so far occur in spatially homogeneous environments where the only spatial inhomogeneities are in the form of differences in signaling among cells. How might our results be influenced by realistic environmental gradients, similar to those generated by abiotic or biotic processes? To answer this, we studied the role of spatial heterogeneity in the evolution of communication by modeling different degrees of intermixing of the environmental states $E_{OFF}$ and $E_{ON}$. In each generation, the environmental state $E_{OFF}$ existed in one of two spatial domains with equal probability, and state $E_{ON}$ existed in the other. These spatial domains were generated using a stochastic spatial pattern generator, where pixels of a grid preferably transition to the state occupied by the majority of their neighbors (Methods). By running this generator for different numbers of steps starting from an initial random configuration of the grid, we were able to generate either fine or coarse-grained domains that were then used as a basis for simulating environments with high and low heterogeneity, respectively (Fig. 4).

When environmental structure is fine-grained, individuals are very likely to interact with neighbors experiencing different environmental regimes. As a result, they are exposed to misleading information about the state of the environment, which in turn impedes the evolution of collective sensing (Fig. 4a). When there is coarse-grained environmental structure, this effect also occurs at the spatial boundaries between environmental states However, cell-cell communication is still beneficial for cells in the center of the spatial domains since they interact with other individuals experiencing the same environmental

conditions. Therefore, in contrast to environments with fine-grained structure, collective sensing evolves more readily when spatial heterogeneity is low (Fig. 4b). Importantly, these findings are contingent on the size of the interaction neighborhood (yellow halo, Fig. 4), which is set by the rate of environmental diffusivity $D$. We illustrate this idea by showing that in the same regime with low levels of spatial heterogeneity, high environmental diffusion can prevent the evolution of communication by increasing the interaction neighborhood of cells (Fig. 4c). High diffusion makes it more likely that any given cell is communicating with others experiencing a different environmental state, eroding the information contained in the external concentration of $A$. Thus, when environments vary spatially, the evolution of collective sensing is also favored if bacteria interact at a local scale.

## The dynamics of signal-negative cheaters

Finally, we asked whether collective sensing would evolve if there is a cost for communication. In the baseline version of our model, where cells must be able to produce autoinducers to be capable of responding to the environment, a cost of communication will only impede the evolution of collective sensing if it exceeds the benefit that bacteria gain from correctly determining the state of the environment (Fig. S10). However, in more complex quorum sensing systems, where the autoinducer and the gene product under the control of quorum sensing are not the same, it is conceivable that costs of communication may induce the evolution of signal-negative cheaters that do not communicate with the rest of the population but benefit from 'listening' to other cells.

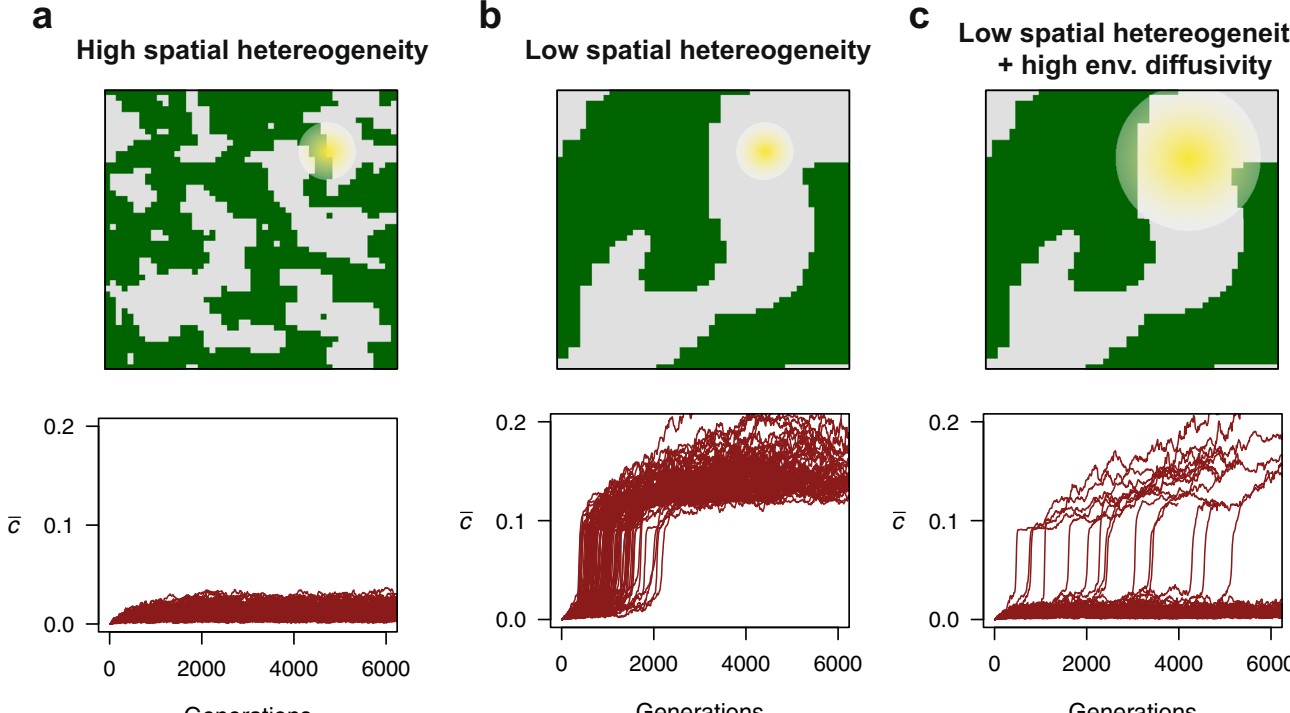

**Fig. 4 | Evolution of quorum sensing in spatially heterogeneous environments.**
**a** (Top) Example of the spatial domains (green vs. gray reflecting $E_{OFF}$ vs. $E_{ON}$ or vice versa) featuring contrasting environmental states in a single evolutionary simulation with high spatial heterogeneity. The yellow halo represents the neighborhood of interaction of a focal cell; the size of this neighborhood depends on the rate of environmental diffusivity $D$. (bottom) Mean $c$ across 4000 generations in 100 replicate evolutionary simulations with high spatial environmental heterogeneity. The evolution of cell-to-cell communication is hindered because communicating cells receive conflicting information from individuals experiencing a different environmental state. Panel (**b**) (top and bottom) shows results for a scenario with low environmental heterogeneity. Cell-to-cell communication evolves in all simulations. Panel (**c**) (top) reflects a scenario with low spatial heterogeneity and a high rate of environmental diffusivity ($D = 50$), as illustrated by the large size of the yellow halo. Despite coarse spatial heterogeneity, high diffusion increases the coupling between cells experiencing different environmental states, generally undermining the information value of the external autoinducer signal. As a result, cell-to-cell communication evolves in only a fraction of the simulations (bottom) and takes on average longer to evolve than in a spatially homogeneous environment. For all panels, unless indicated otherwise, parameters are as in Fig. 2.

To explore the scope for the evolution of such cheaters, we extended our model by incorporating a gene product $P$ whose expression determines fitness and is under the control of $A$ (Fig. S11A). In this way, $A$ acts only as the autoinducer and signal-negative mutants can arise without compromising their ability to express alternative phenotypes in the two states of the environment. Simulations of the extended model show that while signal-negative mutants have an advantage in $E_{OFF}$ environments (i.e. when quorum sensing is not needed), they produce insufficient $P$ in $E_{ON}$ environments, where they rely on $A$ from the extracellular environment to upregulate the expression of $P$ (Fig. S11B). While signal-negative mutants can compensate for this deficiency by increasing the sensitivity of $P$ expression to $A$, their spread is eventually halted by negative frequency-dependent selection (Fig. S11C). This occurs because signal-negative mutants do not contribute to the extracellular concentration of $A$, which undermines the effectiveness of collective sensing in their local neighborhood. Accordingly, the rise of signal-negative mutants is also limited when environmental diffusivity is low, which further contributes to the stability of collective sensing against the evolution of cheating.

## Discussion

Our model shows that quorum sensing could have evolved solely as a result of its collective sensing functionality, without a benefit resulting from the coordinated action among cells. This functionality can explain why several environmental parameters exert a tight control over the rate of autoinducer production across different quorum sensing systems. Moreover, it offers an alternative explanation as to why bacteria use quorum sensing to regulate the expression of

'private' functions which has been previously interpreted as a strategy to stabilize public good-mediated cooperation[9,41]. Importantly, that bacteria engage in collective sensing does not preclude that cells also benefit from coordinated action[6,42] and in principle both mechanisms could operate simultaneously in the same quorum sensing system and be relevant for the regulation of private and public functions, respectively.

We assume a very simple network of gene regulation with a single component $A$ that diffuses passively through the cell membrane and acts both as the autoinducer and the end product under quorum sensing control. In nature quorum sensing architectures are more complex and contain several components that could potentially reflect the state of the environment. For instance, antibiotics trigger competence by upregulating the expression of the entire *com* operon[14,17]. This operon includes genes responsible for the machinery of autoinducer production and export as well as for a histidine kinase and response regulator comprising the autoinducer receptor complex. While other environmental variables can have more targeted effects involving the up or downregulation of only specific components of a quorum sensing system, these effects would have to be reflected in the rate of secretion of autoinducers into the extracellular space for collective sensing to work.

Two basic features of bacterial interaction networks could have facilitated the evolution of cell-cell communication for collective environmental sensing. First, when a bacterium divides, its daughter cell often remains close in space. This feature not only protects quorum sensing from cheater invasion via a kin selection mechanism[43,44], but as shown here also facilitates the emergence of

sufficiently large clusters of communicators for collective sensing to be beneficial. Second, bacterial interactions occur over short spatial ranges, on the order of microns for certain quorum sensing systems as reported recently[45–47]. We show here that this feature of bacterial communication could have favored the evolution of collective sensing because emergent communicators (i) are often coupled to their communicating offspring (Fig. 3) and (ii) avoid long-range interactions with cells located in different microenvironments that could share deceptive information (Fig. 4).

One of the main obstacles in explaining the evolution of quorum sensing is how this process remains stable in the presence of cheaters. In the context of collective sensing, mutants that do not respond to autoinducers would not have a fitness benefit because private functions cannot be outsourced to neighboring cells. However, collective sensing could be prone to the evolution of signal-negative mutants that cheat and do not share information but only listen to others. We found that these mutants have a disadvantage in environments that require upregulation of quorum sensing because they rely exclusively on autoinducers produced by other cells which might not be sufficient for full or timely quorum sensing activation. Moreover, even if these cheaters were to compensate for such deficiency by becoming more sensitive to autoinducers, they are subject to negative-frequency dependent selection and have a limited advantage when there is low diffusivity of autoinducers in the extracellular space (Fig. S11). Since the cost of erroneous decisions regarding the state of the environment is likely higher than the cost of producing autoinducers, we expect that these different mechanisms will limit the evolution of signal-negative cheaters. Importantly, this further emphasizes that the local nature of bacterial interactions could have favored the evolution of collective sensing and is consistent with previous observations from several quorum sensing systems[48,49].

Collective sensing has been proposed as a mechanism for decision-making in other systems resulting from social interactions among individuals with simple behavioral rules[21,50]. A notable instance in several animal species is "the wisdom of crowds", where individuals can improve estimation accuracy by aggregating their separate estimates of the environmental state. Examples of this phenomenon range from nest-site choices by ant colonies[51,52], to foraging decisions by fish schools[53] and even medical diagnostics in humans[54,55]. Although bacteria do not possess the complex sensing, cognition and feedback mechanisms found in social animals and humans, our work shows that the same collective functionality can arise with simple gene regulation networks and could have driven the evolution of quorum sensing as one of the most widely used communication systems in bacteria.

## Methods

We study the evolution of cell-to-cell communication in a bacterial population encountering varying environments. The phenotype of a cell is determined by a simple network of positive regulation where a gene product $A$ promotes its own transcription (Fig. S1). We model this positive feedback by assuming that the transcriptional regulation of $A$ follows standard Hill kinetics. Bacterial cells inhabit a two-dimensional grid of size $N \times N$ where they can communicate with other cells by exchanging $A$ with the extracellular space. $A$ is exchanged by passive diffusion with a diffusion constant $c$. Based on these assumptions the system of equations describing the intracellular concentration of $A$ and extracellular concentration, $A_E$, in a single grid space is

$$\frac{dA}{dt} = \frac{k_0 + k(A/K)^n}{1 + (A/K)^n} + c(A_E - A) - dA \quad (1)$$

$$\frac{dA_E}{dt} = \alpha c(A - A_E) + D\nabla^2 A_E \quad (2)$$

where $k_O$ is the basal rate of $A$ production when the promoter is not bound to any molecule of $A$, $k$ is the maximal production rate, $K$ is the dissociation constant, $n$ is the degree of cooperative binding, $d$ is the rate of degradation of $A$, $\alpha$ is the ratio between the intra and extracellular volume and $D$ is the rate of diffusion of $A$ in the extracellular space. Besides passive diffusion through the membrane, the intracellular concentration of $A$ is determined by a production term that reflects cooperative binding of $A$ (which is characteristic of many autoinducers) and a degradation term that accounts for factors such as dilution of autoinducers due to cell growth and intracellular enzymatic degradation. Since most QS systems exhibit bistability we choose parameter values that result in two stable states at a high and low concentration of $A$ when $c = 0$ (Fig. S1). These parameter values are $n = 3$, $K = 50$, $d = 0.3$, $k_O = 2.9$ and $k = 29$. Every generation we solve the previous system of equations for a fixed number of time steps $T = 100$ and calculate fitness at the end to determine which individuals will leave offspring in the next generation. We model external diffusion of the autoinducer by applying a gaussian diffusion kernel over the grid containing the values of $A_E$ and assume periodic boundary conditions. The default values for $D$ and $\alpha$ are $D = 0.5$ and $\alpha = 1$. However, since these parameters govern the extracellular dynamics of $A$, we vary them throughout the manuscript to study the role of local interactions on the evolution of collective sensing (see Figs. 3, 4, S8, S9 and S11).

### Fitness calculation and reproduction

In every generation a bacterial population faces one of two possible environments with equal probability. Each environment has an optimal expression level of $A$, denoted by $A_{OFF}$ or $A_{ON}$. $A_{OFF}$ and $A_{ON}$ are set at the stable equilibria of the bistable system when $c = 0$ (Fig. S1). At the start of a generation all cells sample their initial intracellular value of $A$ from a truncated normal distribution with mean either $A_{OFF}$ or $A_{ON}$ depending on the environment and standard deviation $\sigma_{OFF}$ or $\sigma_{ON}$. The fitness of a cell is determined by how well its intracellular $A$ concentration matches the state of the environment throughout the duration of a generation. The fitness function is,

$$w(\triangle_A) = \frac{1}{1 + e^{s(\triangle_A - x)}} \quad (3)$$

where $s$ determines the strength of selection, $\triangle_A = \frac{1}{T}\sum_{t=1}^{T}|A_t - A_{ON}|$ (i.e. the average difference over the $T$ time steps between $A$ and the optimal level of $A$ expression in the current environmental state, in this example $E_{ON}$) and $x$ is the the midpoint of the sigmoid curve. In all simulations we set $s = 0.8$ and $x = 20$. For this choice of parameters $w$ has a sigmoidal shape that strongly penalizes cells that are in the non-optimal phenotypic state but not cells that slightly deviate from the optimal expression levels (Fig. S2). Lower values of $s$ result in flatter fitness functions that impose a lower selective pressure on correctly determining the state of the environment. As a result, collective sensing becomes less profitable with lower $s$ (see Fig. S5).

At the end of every environmental cycle the fitness of each cell is calculated and the entire grid is repopulated by sampling with replacement $N$x$N$ individuals. The probability that an individual is chosen for reproduction equals to its fitness normalized by the total fitness of the population. Upon reproduction, the algorithm for creating and placing the offspring of a cell in the new grid is the following,

1. Draw a random number to determine if $c$ mutates. If $c$ mutates, draw an additional random number to determine whether the new value of $c$ is $c + \partial$ or $c - \partial$. If $c - \partial < 0$, $c$ does not mutate.
2. Calculate the euclidean distance of the mother cell to all other cells in the grid.
3. Place the new cell in the closest grid cell to the mother cell that is still empty.

This algorithm is applied to the $NxN$ vector containing the coordinates of all the cells that will reproduce and it ensures that the offspring of a cell remains close to the location of its mother cell. In simulations where the offspring of a cell is randomly placed on the two dimensional grid, the grid is filled by rows in the order that cells appear in the $NxN$ vector.

In order to model the evolution of collective sensing when communication is costly (see Fig. S10), we assume that the fitness function is given by,

$$w(\triangle_A)e^{-\gamma c} \qquad (4)$$

where $\gamma \geq 0$ sets the extent by which fitness is reduced by $c$. Thus, while $\gamma = 0$ corresponds to the original model where communication is not costly, for values of $\gamma > 0$ cells are increasingly penalized for having higher values of $c$.

## Spatial heterogeneity

We model spatial variation by using an Ising model[56] to establish the initial configuration of the environment. Using this model we can vary the scale of spatial heterogeneity from a random configuration to a homogeneous grid. In two dimensions, this model consists of a grid where cells can be in two possible states (−1 or +1). The total energy of the system is determined by whether neighboring cells are in the same or in a different state and is given by,

$$E = -J\sum_{<ij>}s_i s_j \qquad (5)$$

where $<ij>$ denotes all the pairs of neighboring cells, $s_x$ is the state of the grid cell $x$ and $J$ determines the sign of the interaction. We assume that $J > 0$ so over time the system converges from a random configuration to a configuration where all the cells have the same state.

Starting from a random configuration where each grid cell is assigned to either of the two states with equal probability, we simulated this model using a Metropolis Monte Carlo algorithm for $I$ number of iterations[57]. Briefly, at each iteration a grid cell is selected at random and its state is flipped. If the energy of the new configuration is lower than the energy of the old configuration the change in state is accepted and the state of the cell is flipped. If $\Delta E \geq 0$, the state of the cell can still be flipped with probability $e^{-\Delta E/S_T}$, where $S_T$ is a scaling constant which is set to $S_T = 0.1$ in all the simulations. In each iteration, this is repeated $NxN$ times. Over time, the grid configuration will become more homogeneous until eventually all the grid cells are in the same state after many iterations.

For each simulation run, we first determine the spatial configuration of the $NxN$ grid by running the previous algorithm for a fixed number of $I$ iterations. By increasing $I$ we can vary the degree of spatial heterogeneity in the model from high ($I = 5$) to low heterogeneity ($I = 55$) as done in Fig. 4. We use the resulting grid configuration composed of the two states, −1 and +1, to determine the state of the environment in each grid cell every generation. At the start of each generation, a random number is drawn to assign the environmental states to the grid states. $E_{OFF}$ and $E_{ON}$ are assigned to grid cells −1 and +1 or vice versa with equal probability every generation. Each bacterial cell samples an initial intracellular $A$ concentration from a distribution whose mean is determined by the environmental state in the grid cell inhabited by the bacterium.

## Signal-negative mutants

We extended our model to study the dynamics of signal-negative mutants by assuming that $A$ controls the expression of a product $P$ that determines fitness (Fig. S11). We assume that the production/degradation dynamics of $P$ is the same as of $A$ so fitness is calculated in the same way as in the original model by comparing the internal concentration of $P$ to an optimal value given by the current environmental state. $P$ does not diffuse through the cell membrane, so its dynamics is governed by the following equation:

$$\frac{dP}{dt} = \frac{k_0 + k(A/K)^n}{1 + (A/K)^n} - dP \qquad (6)$$

For cooperative cells that produce and sense the autoinducer $A$, the equations for $A$ and $A_E$ are (1) and (2). For signal-negative mutants, the equation for $A_E$ is (2) but there is no production of $A$. Thus, the concentration of $A$ in these cells is only affected by diffusion through the membrane and intracellular degradation, such that its dynamics is given by

$$\frac{dA}{dt} = c(A_E - A) - dA \qquad (7)$$

Finally, we assume that the fitness of cooperative cells is reduced by a fixed factor due to the production of $A$. Since we study the fitness advantage of signal-negative mutants emerging in a population of cooperative cells, we refer to the latter as wildtype (see Fig. S11).

## Reporting summary

Further information on research design is available in the Nature Portfolio Reporting Summary linked to this article.

## Data availability

All data can be generated by running the available code.

## Code availability

The Python code used for the simulations is available at Zenodo (https://doi.org/10.5281/zenodo.7799906).

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

## Acknowledgements
We thank Martin Ackermann, Franz J. Weissing and Simon van Vliet for comments on earlier versions of this manuscript. SMG and GSvD were supported by a Starting Independent Researcher Grant (309555) of the European Research Council and a Vidi fellowship (864.11.012) of the Netherlands Organization for Scientific Research. SMG was supported by a James S. McDonnell Foundation 21st Century Science Initiative Understanding Dynamic and Multi-scale Systems Postdoctoral Fellowship Award (2020-1456) and a Rubicon grant (019.201EN.041) from the Netherlands Organization for Scientific Research.

## Author contributions
SMG, MEH and GSvD concived the project. SMG developed the model and analysed the data with feedback from MEH and GSvD. SMG wrote the paper with feedback from MEH and GSvD.

## Competing interests
The authors declare no competing interests.
