## [Peer Review File · Nature Communications]

Quorum sensing as a mechanism to harness the wisdom of the crowdsReviewers' Comments:

Reviewer #1:

Remarks to the Author:

Review to "Quorum sensing as a mechanism to harness the wisdom of the crowds" by Stefany Moreno-Gamez, Michael E. Hochberg, G.S. van Doorn

Short summary:

The paper considers the general concept of Quorum sensing (QS) with the perspective of a collective behaviour, but more from an exchange of information perspective about the environmental condition. General ideas are transferred to a model approach, not with the aim of considering realistic situations but to underline the general concept.

General Comment:

The paper is well-written and thoroughly prepared, the figures are helpful and also thoroughly prepared. The general idea of the manuscript is of current interest and for sure worth to consider, to improve understanding of QS and the different levels where the release and sensing of the signal molecules might play a role. The idea in general, e.g. the pooling of imperfect estimates by "bringing together" what single cells "sense" or "observe", sounds good. Even though this aspect is not completely new, understanding could and should still be improved. So far I fully agree with the authors.

It becomes more critical when going into some details. Unfortunately, some important aspects are completely left out (or only mentioned quickly, without taking it into account e.g. for the model approach).

The authors focus on estimating the environmental conditions, which is fair and interesting. I'm missing a thorough discussion, how separated this aspect can be considered as all, as also here, the cells automatically get information about the presence of neighbouring cells – even more "blurred" due to the positive feedback loop included and the resulting bistability. I would say, it's hard up to impossible to distinguish this "population information" from the "environmental information", and is not taken at all into the modelling approach (e.g. by checking how the situation changes, if stronger or weaker upregulation takes place etc.). The bacteria cannot avoid to receive individual information, environmental information, population information, by using just one signal (as assumed in the approach here).

As second critical point concerns that cheaters are more or less left out from the discussion, but they may be the most essential "problem" to understand how QS could "survive".

Few concrete points to mention:

- l.54: It remains unclear to me on which basis the authors come to the conclusion that private goods/functions are more important for individual fitness than public goods. E.g. thinking about siderophores, without iron no growth possible at all, but to my understanding that would concern a public good. Furthermore, the authors mention themselves (at least very briefly) that bacteria do NOT only monitor cell density (e.g. follow the discussion about diffusion sensing, or with a more general viewpoint efficiency sensing). I think this statement is not correct in general and should be re-formulated.
- l.68-71: I'm missing a bit more discussion about the new aspects of interpretation of the "QS" compared to the available "integrative perspectives" (as the authors call it), also efficiency sensing. Even more, they seem to ignore it later a bit, by often separating something like "sensing environmental conditions (collectively)" versus "sensing population density" – where it's already standard that obviously the cells cannot separate, they always get a mixture of information of the different aspects.
- Fig. 1: The model looks a bit artificial to me, assuming a fine grid with just one bacterial cell in grid point, which leads in the extreme case to environmental conditions changing every micrometer ... difficult to imagine (unless they all live in little wholes for their own, but probably that's not the purpose of the authors, as this would clearly be the diffusion sensing situation).

- L.125ff: I was wondering what exactly means "the fittest individuals", how strict it is. I have seen the sigmoidal function, but where is it cut then to decide which cells divide and which don't. In a realistic setting, also cells with less fitness could still divide. Furthermore, this process of choice represents another positive feedback including modelling artefacts, which is not addressed in detail, any more detailed discussion welcome.

- Fig. 1 <-> Table 1: The rough structure of the gene regulatory network looks like that of a Gram-negative bacterium, whereas in Table 1, many examples for Gram-positive bacteria are mentioned. Maybe better to adapt, to avoid an inconsistent impression (and to formulate more clearly, on what the focus is set on).

- L. 171 ff.: It seems like the state of the environment is reflected directly by A (as also the model shown in the supplementary material just considers A, intracellular and extracellular). I'm missing at least a short discussion, why A is the adequate player to do this, and not e.g. the autoinducer-receptor complex, which doesn't change instantaneously, better and more realistically takes into account the "history" and the bistability – all of these may play an important role for the decision of the bacteria. A comment on that would be very helpful.

- L.185ff.: The authors consider cells which share information or not by considering the parameter c. There isn't any clear discussion about what $c=0$ would mean – not only stopping sharing information (communication) with other bacteria, but also stopping e.g. "diffusion sensing" for the single, individual cell, and by that any benefit from that (this appears again on line 242 and should be clarified also there). This makes me critical of thinking if c was really the adequate approach to consider that. Very critically said:

If $c=0$ (or a very, very small c) would be the outcome for being beneficial for the bacterial cells, that would mean, the whole system is completely worthless and only waste of energy. Thus, it should be expected anyway, that some $c>0$ is better. And is the result not already founded in the structure of the model?

- In the same context: I see that the authors do not intend to consider realistic parameter values with real world units, which is of course ok for a general setting. Nevertheless, one should not completely forget about reality, e.g. by comparing the order of magnitude of c (or the outcome which c would be best for the bacteria in the simulations) with free diffusion – at least c shouldn't become faster than that. A comment on that would be very helpful.

- L.220ff.: I think it's difficult to formulate a sentence like that. Maybe there is no benefit of cell-to-cell communication for improving individual estimates of environmental conditions. But that is for sure not the only purpose, as seen in Table 1 and many other publications, there are so many different purposes of the QS mechanism, so it's completely fine if the main benefit for cell-to-cell is more focussed on the local cell density aspect or so.

Additional questions: What is meant by "high c"? Compared to what?

- L. 330ff.: I do not fully agree with this general statement, as the whole mechanisms would work even with single cells, completely without exchange, as could be seen at many occasions. This needs to be clarified.

- L. 347 ff.: But this has not much to do with the transporting mechanism (versus free diffusion). It's the general problem the cells may have that they produce molecules which (which mechanism ever) get outside and others react on that. Cheater could still "listen", but save energy i.e. not producing by themselves.

Thus, it would be essential to take exactly such cheaters into account, in the model setup and then to show that these cheaters have a disadvantage. That would really help for understanding. Here it's "only" shown that QS can develop, but under the (quite restricting) assumption that there are no cheater which use the information of the others without contributing by themselves.

Concerning the Supplementary text:

- A is not a protein (at least not usually for Gram-negative bacteria)

- I was wondering why the authors let the basal production decrease for large A? Most "classical models" in literature keep it constant and only have the upregulated production (with the positive feedback) dependent on A. This should be adapted or explained, why this modified term is necessary.

- A and A_E are mentioned to be concentrations. By that I can understand the terms $(A_E - A)$ in the model for the exchange, but to have no conversion factor (for the extracellular versus the intracellular

volume) included, obviously means that the same volume is assumed for both, which is not realistic. This needs to be corrected.

- Why there is no abiotic degradation of the extracellular A_E included (but for the intracellular A it is)? This needs to be added or explained.
- At least a rough explanation how the parameter values were chosen would be highly appreciated. Does the model behave similarly when other parameter values are applied? E.g. the Hill coefficient seems to be very high with $n=6.75$ (for many typical bacteria it's 2 (having the dimers) or maybe something between 2 or 3). Or k_0 and k only differ by a factor of approx. 5, quite low, often the increased production in bacterial species is observed to be something between 10 and 100.
- I didn't fully understand: when placing the cells for the next generation: can there cells stay empty, and if yes, how many are that typically? Making this easier to understand / to read would be appreciated.

Reviewer #2:

Remarks to the Author:

In this paper, the authors develop an evolutionary model to show that quorum sensing can evolve for the function of improving information about the environment when individual information is noisy. Specifically, the authors find that quorum sensing readily evolves when the diffusivity of the extracellular environment is intermediate and environmental heterogeneity is not too small-scaled. I find the model to be concisely tailored to investigate this phenomenon, the paper to be clearly written, and consider the results to be interesting and potentially impactful. I do have some relatively minor concerns, which I outline below. If these concerns are addressed, I think the paper would make a valuable contribution to the literature.

1. Explanation of timings or environmental cycling/reproduction in the model (lines 109-21). It is currently not entirely clear what constitutes an 'environmental cycle' in the model and exactly at what point reproduction takes place (and thus with what frequency reproduction occurs relative to the environmental fluctuations). There is both mention of 'non-overlapping generations' (line 126) and of 'bud(ding) multiple times during an environmental cycle' (line 120). Perhaps this could be explained in somewhat more explicit terms.
2. Fitness function (line 122 and Supplementary Info). The authors assume a sigmoidal shape of the fitness function so that cells are only strongly penalized for misdetermining whether the environment is in the ON or OFF state but not for small deviations. How did the authors choose the steepness of this function?
3. Constraints on variable c (lines 127-8 and Supplementary Info). It is mentioned that c is constrained to be equal or larger than 0. Should it not also be constrained to be smaller or equal than 1?
4. Initial concentration of A (lines 161-2). The authors mention that they draw the initial concentration of A from a uniform distribution of which they specify only the mean. It would be more complete to just give the range of this distribution (based on only the mean we cannot derive this range).
5. Cost of permeability (lines 347-8). The authors talk about their assumption that they assume passive diffusion of A across the membrane, but that a cost of permeability (in case of active secretion and sensation) might lead to different results. Do the authors have any ideas on how changes in passive permeability could evolve, as is assumed in their model? If this would occur through changes in the molecular structure of A, this might then need to be accompanied by mutations in the receptors for A as well. It is easier to see how active secretion and sensation could evolve relatively continuously, but in this case (as the authors note) it would make sense to implement a cost associated with these functions. I think this should perhaps be fleshed out a bit more. Perhaps the

authors could explain to what extent their assumptions around this are meant to be realistic, and if they are not meant to be very realistic, how they would expect more realistic assumptions (e.g. involving costs) would change the outcome of the model. Would the system completely collapse because of cheating or would they still expect that there are conditions in which communication could evolve?

REVIEWER COMMENTS

Reviewer #1 (Remarks to the Author):

Review to "Quorum sensing as a mechanism to harness the wisdom of the crowds" by Stefany Moreno-Gamez, Michael E. Hochberg, G.S. van Doorn

Short summary:

The paper considers the general concept of Quorum sensing (QS) with the perspective of a collective behaviour, but more from an exchange of information perspective about the environmental condition. General ideas are transferred to a model approach, not with the aim of considering realistic situations but to underline the general concept.

General Comment:

The paper is well-written and thoroughly prepared, the figures are helpful and also thoroughly prepared. The general idea of the manuscript is of current interest and for sure worth to consider, to improve understanding of QS and the different levels where the release and sensing of the signal molecules might play a role. The idea in general, e.g. the pooling of imperfect estimates by "bringing together" what single cells "sense" or "observe", sounds good. Even though this aspect is not completely new, understanding could and should still be improved. So far I fully agree with the authors. It becomes more critical when going into some details. Unfortunately, some important aspects are completely left out (or only mentioned quickly, without taking it into account e.g. for the model approach).

The authors focus on estimating the environmental conditions, which is fair and interesting. I'm missing a thorough discussion, how separated this aspect can be considered as all, as also here, the cells automatically get information about the presence of neighbouring cells – even more "blurred" due to the positive feedback loop included and the resulting bistability. I would say, it's hard up to impossible to distinguish this "population information" from the "environmental information", and is not taken at all into the modelling approach (e.g. by checking how the situation changes, if stronger or weaker upregulation takes place etc.). The bacteria cannot avoid to receive individual information, environmental information, population information, by using just one signal (as assumed in the approach here).

As second critical point concerns that cheaters are more or less left out from the discussion, but they may be the most essential "problem" to understand how QS could "survive".

We thank the reviewer for the appreciation of our manuscript and for all the helpful comments.

We fully agree that in nature bacteria might benefit from using the concentration of AIs to estimate 'population information' as well as 'environmental information' and thus might need to disentangle these sources. In our model we intentionally assume that cells only benefit from estimating information about the state of the environment and not from estimating

population-level parameters (e.g. population density). In particular, we assume that population size is constant every generation and that a cell is always surrounded by the same number of neighbors. We made this choice because we wanted to focus on characterizing the role of collective sensing as a novel mechanism to explain the evolution of QS, but also because the question of how bacteria use different types of information has been addressed by previous models (e.g. Cornfoth et al 2014). We now clarify these points in the text. See lines 131-135:

“... Importantly, we assume that matching the state of the environment is the only factor that determines the fitness of a cell. In nature, bacteria might benefit from estimating other quantities from the concentration of AIs in addition to the state of the environment (e.g. population density), but we purposely left these aside to focus on the role of collective sensing on driving the evolution of QS.”

Also, see lines 137-144:

“...Reproduction occurs at the end of each environmental cycle and cells are selected to reproduce with a probability proportional to their fitness. Reproducing cells are sampled with replacement, and offspring are collected until their total number is sufficient to replace the parental generation and fully repopulate the grid. As a consequence, population size remains constant, generations are non-overlapping and individuals can have multiple descendants in the next generation.”

Regarding the presence of cheaters, it is well established that there are two potential ‘cheating’ mechanisms in the context of QS: Bacteria can ‘cheat’ if they do not respond to AIs and avoid producing molecules under QS control or if they avoid AI production but respond to AIs produced by others. Since collective sensing is based on the regulation of private functions that cells cannot outsource to others, we do not expect the former type of cheaters to have a fitness advantage. However, mutants that do not produce AIs could potentially emerge in the context of collective sensing.

In the original version of our model signal-negative mutants cannot arise even if communication is costly because A acts as the autoinducer and the product under QS control that determines fitness. For this reason we have now extended our model to study the dynamics of these mutants by decoupling these two functions of A. We dedicate a full section of the Supplementary Materials to explain this model extension and describe the results we obtained in lines 357-378:

“Finally, we asked whether collective sensing would evolve if there is a cost for communication. In the baseline version of our model, where cells must be able to produce AIs to be capable of responding to the environment, a cost of communication will only impede the evolution of collective sensing if it exceeds the benefit that bacteria gain from correctly determining the state of the environment (Figure S10). However, in more complex QS systems, where the autoinducer and the gene product under the control of QS are not the same, it is conceivable that costs of

communication may induce the evolution of signal-negative cheaters that do not communicate with the rest of the population but benefit from 'listening' to other cells.

To explore the scope for the evolution of such cheaters, we extended our model by incorporating a gene product P whose expression determines fitness and is under the control of A (Figure S11A). In this way, A acts only as the autoinducer and signal-negative mutants can arise without compromising their ability to express alternative phenotypes in the two states of the environment. Simulations of the extended model show that while signal-negative mutants have an advantage in E_{OFF} environments (i.e. when quorum sensing is not needed), they produce insufficient P in E_{ON} environments, where they rely on A from the extracellular environment to upregulate the expression of P (Figure S11B). While signal-negative mutants can compensate for this deficiency by increasing the sensitivity of P expression to A , their spread is eventually halted by negative frequency-dependent selection (Figure S11C). This occurs because signal-negative mutants do not contribute to the extracellular concentration of A , which undermines the effectiveness of collective sensing in their local neighborhood. Accordingly, the rise of signal-negative mutants is also limited when environmental diffusivity is low, which further contributes to the stability of collective sensing against the evolution of cheating.”

Based on these findings, we also rewrote the Discussion section referring to cheaters in the context of collective sensing. See lines 412-426:

“One of the main obstacles in explaining the evolution of QS is how this process remains stable in the presence of cheaters. In the context of collective sensing, mutants that do not respond to AIs would not have a fitness benefit because private functions cannot be outsourced to neighboring cells. However, collective sensing could be prone to the evolution of signal-negative mutants that cheat and do not share information but only listen to others. We found that these mutants have a disadvantage in environments that require upregulation of quorum sensing because they rely exclusively on AIs produced by other cells which might not be sufficient for full or timely QS activation. Moreover, even if these cheaters were to compensate for such deficiency by becoming more sensitive to AIs, they are subject to negative-frequency dependent selection and have a limited advantage when there is low diffusivity of AIs in the extracellular space (Figure S11). Since the cost of erroneous decisions regarding the state of the environment is likely higher than the cost of producing AIs, we expect that these different mechanisms will limit the evolution of signal-negative cheaters. Importantly, this further emphasizes that the local nature of bacterial interactions could have favored the evolution of collective sensing and is consistent with previous observations from several QS systems (58, 59).”

Few concrete points to mention:

- **I.54: It remains unclear to me on which basis the authors come to the conclusion that private goods/functions are more important for individual fitness than public goods. E.g. thinking about siderophores, without iron no growth possible at all, but to my understanding that would concern a public good. Furthermore, the authors mention**

themselves (at least very briefly) that bacteria do NOT only monitor cell density (e.g. follow the discussion about diffusion sensing, or with a more general viewpoint efficiency sensing). I think this statement is not correct in general and should be re-formulated.

We fully agree that for several QS systems (e.g. siderophore production) the regulation of public goods is much more relevant for individual fitness than the regulation of private functions. Our statement was meant to refer to systems where the opposite might be true. We have now rewritten this part to clarify this point (see lines 49-56):

“...There is evidence of this idea in systems where QS controls the production of ‘public goods’ (e.g. extracellular proteases). In this context, secreting costly molecules is more efficient if other cells engage in the same behavior and thus the benefit of upregulating these traits increases with the number of cells (6). However, in other systems QS primarily regulates the expression of ‘private’ functions such as competence or persistence that are not shared with other individuals (3, 7–9). Since private functions -unlike public goods- can be beneficial for a cell regardless of the number of neighboring cells expressing them, it is less clear why bacteria should regulate these functions by monitoring population density.”

• I.68-71: I'm missing a bit more discussion about the new aspects of interpretation of the "QS" compared to the available "integrative perspectives" (as the authors call it), also efficiency sensing. Even more, they seem to ignore it later a bit, by often separating something like "sensing environmental conditions (collectively)" versus "sensing population density" – where it's already standard that obviously the cells cannot separate, they always get a mixture of information of the different aspects.

We thank the reviewer for pointing out that we are missing some discussion on efficiency sensing. We now refer specifically to this hypothesis. Although we agree bacteria cannot easily disentangle cell density from environmental diffusivity, we think efficiency sensing still overemphasizes diffusion as a major driver of QS evolution. See lines 64-71:

“...These and other observations led to the ‘diffusion sensing’ hypothesis, which states that bacteria release AIs to test environmental diffusivity and regulate the secretion of costly molecules into the extracellular environment (11). This hypothesis was later reformulated as ‘efficiency sensing’ to acknowledge that bacteria cannot disentangle local cell density from environmental diffusivity using the concentration of AIs but instead rely on both factors to determine the efficiency of producing costly diffusible molecules. Nevertheless, emphasizing diffusion as the main functional driver of QS likely underestimates the complexity of QS regulation given that many other factors such as pH, oxygen and antibiotic stress can influence QS as well (12–14).”

Also, note that in the context of our hypothesis, bacteria would not gain much information from accurately estimating population density itself, which might indeed be difficult if many other

factors simultaneously affect the AI concentration. Instead, we are suggesting that the product of cell density * rate of AI production by single cells is more useful to bacteria. This product is determined by how individual cells sense several environmental factors and would reflect their 'assessment' of whether it is beneficial to switch on QS in their current environment. Therefore this product can be high even when the actual population density is low if bacteria are facing an environment where upregulating QS-controlled traits is beneficial.

In this sense, our hypothesis is not different from the available 'integrative perspective' in terms of what information cells are sensing. Instead, what differs between the two is the functional aspect: in collective sensing, the functional value lies in being able to estimate the environmental state more reliably, whereas, in the traditional view, the functional value lies in coordinated action.

• Fig. 1: The model looks a bit artificial to me, assuming a fine grid with just one bacterial cell in grid point, which leads in the extreme case to environmental conditions changing every micrometer ... difficult to imagine (unless they all live in little wholes for their own, but probably that's not the purpose of the authors, as this would clearly be the diffusion sensing situation.

We agree with the reviewer that in simulations involving spatial structure it would be more realistic to have a gradual rather than a sharp demarcation at the boundaries between different environmental conditions. We expect nevertheless that any effects would be quantitative and not affect the qualitative conclusions of the simulations we present in Figure 4: If cells communicate over very long ranges where the environment changes considerably this would impede the evolution of collective sensing.

• I.125ff: I was wondering what exactly means "the fittest individuals", how strict it is. I have seen the sigmoidal function, but where is it cut then to decide which cells divide and which don't. In a realistic setting, also cells with less fitness could still divide. Furthermore, this process of choice represents another positive feedback including modelling artefacts, which is not addressed in detail, any more detailed discussion welcome.

We have clarified the description of this part of our model. Individuals are not selected for reproduction in a deterministic manner but instead are chosen with a probability proportional to their fitness. See lines 137-140:

"Reproduction occurs at the end of each environmental cycle and cells are selected to reproduce with a probability proportional to their fitness. Reproducing cells are sampled with replacement, and offspring are collected until their total number is sufficient to replace the parental generation and fully repopulate the grid. ..."

We have also clarified this in the figure legend. See lines 176-179:

“(D) Fitness values are used to update the grid for the next environmental cycle. The grid is repopulated such that every individual has the chance of reproducing with a probability proportional to its fitness and its descendants are placed at or in adjacent locations to its position in the grid.”

Also, note that the steepness of the fitness function is what determines the strength of selection in our simulations: If the fitness function would be totally flat, then there would be no fitness differences between cells that make correct or incorrect estimates of the environment and collective sensing would not evolve. As the steepness of the fitness function increases the incentive to communicate becomes higher. We now discuss this issue in the supplementary materials and have included a supplementary figure (Figure S5) illustrating how different parameterizations of the fitness function affect the benefit gained by collective sensing.

• Fig. 1 <-> Table 1: The rough structure of the gene regulatory network looks like that of a Gram-negative bacterium, whereas in Table 1, many examples for Gram-positive bacteria are mentioned. Maybe better to adapt, to avoid an inconsistent impression (and to formulate more clearly, on what the focus is set on).

Thanks for pointing this out. We have now added citations to examples of the different phenotypes we mention in Gram negatives (e.g. persistence in *A. baumannii* and metabolic rewiring in *Y. pestis*, *B. glumae* and *P. aeruginosa*).

• L. 171 ff.: It seems like the state of the environment is reflected directly by A (as also the model shown in the supplementary material just considers A, intracellular and extracellular). I'm missing at least a short discussion, why A is the adequate player to do this, and not e.g. the autoinducer-receptor complex, which doesn't change instantaneously, better and more realistically takes into account the "history" and the bistability – all of these may play an important role for the decision of the bacteria. A comment on that would be very helpful.

Given the simple structure of our model and our assumptions that diffusion is passive and A acts both as autoinducer and QS product, there are no other potential players of the QS network that could reflect the state of the environment in our model. Although these assumptions restrict the type of dynamics we can study (e.g. the evolution of cheating), we deliberately chose such a simple architecture to focus on characterizing the basic requirements needed for the evolution of collective sensing.

This said, for collective sensing to evolve even in a more complex model, the state of the environment would have to be reflected in the rate of autoinducer secretion. This could happen through other QS players that reflect the state of the environment and in turn influence the production of AIs (e.g. more receptors will lead a cell to sense and produce more autoinducers).

Nevertheless, the mechanism we are proposing is based on cells sharing information about the state of the environment and the only component of a QS system that permits this are autoinducers. Thus, since autoinducer secretion is the way bacteria can 'cast' their votes to collectively decide what to do, the estimate that each cell makes of its environment should at the very least be reflected in this QS component.

We now discussed this point more clearly in lines 389-398:

"We assume a very simple network of gene regulation with a single component A that diffuses passively through the cell membrane and acts both as the autoinducer and the end product under QS control. In nature QS architectures are more complex and contain several components that could potentially reflect the state of the environment. For instance, antibiotics trigger competence by upregulating the expression of the entire *com* operon (14, 17). This operon includes genes responsible for the machinery of AI production and export as well as for a histidine kinase and response regulator comprising the AI receptor complex. While other environmental variables can have more targeted effects involving the up or downregulation of only specific components of a QS system, these effects would have to be reflected in the rate of secretion of AIs into the extracellular space for collective sensing to work."

• L.185ff.: The authors consider cells which share information or not by considering the parameter c . There isn't any clear discussion about what $c=0$ would mean – not only stopping sharing information (communication) with other bacteria, but also stopping e.g. "diffusion sensing" for the single, individual cell, and by that any benefit from that (this appears again on line 242 and should be clarified also there). This makes me critical of thinking if c was really the adequate approach to consider that. Very critically said: If $c=0$ (or a very, very small c) would be the outcome for being beneficial for the bacterial cells, that would mean, the whole system is completely worthless and only waste of energy. Thus, it should be expected anyway, that some $c>0$ is better. And is the result not already founded in the structure of the model?

We agree with the reviewer that the description of c was not clear in the main text. We have now rewritten this part in the section where we describe the model, together with a brief description of the scenario where $c=0$. See lines 108-114:

"In order to study how communication evolves in our model, we let bacteria evolve a parameter c that sets the rate of passive diffusion of A through the cell membrane. This parameter determines the membrane permeability to A and thus the degree of communication between cells (e.g. when $c = 0$, a cell does not share or receive any information from other cells). In nature, membrane permeability depends in part on the biochemical properties of AIs and can change because of variations in AI length or molecular structure, as well as by the evolution of active mechanisms for AI secretion or transport (e.g. carrier proteins) (28–30)."

Note that in our model cells do not gain any benefit from diffusion sensing since we don't assume that bacteria secrete costly molecules to the extracellular space (fitness is only

determined by the internal concentration of A). That said, when developing the model we were also concerned that increasing c could be beneficial for bacteria even in the absence of communication – this would happen if cells could gain information on the environmental state from the initial concentration of A in the extracellular space. To avoid this artifact and a bias towards the evolution of high c , the extracellular concentration of A at the start of every environmental cycle is sampled from a uniform distribution, which makes this concentration uninformative of the current state of the environment (see lines 146-153):

“Exchanging A with the extracellular environment provides cells with information on the initial extracellular concentration of A, which could potentially be beneficial if this concentration is informative of the current environmental state. To prevent such benefits (which do not result from cell-cell communication) from biasing the outcome towards the evolution of high c , we implement initial conditions for the extracellular concentration of A that are uninformative to cells. In particular, we assume that, for each generation of cells, the initial concentration of A is sampled independently for each grid cell from a uniform distribution in the interval $[0, A_{OFF} + A_{ON}]$.”

We confirmed that increasing c is beneficial for bacteria only if they can communicate with others. We did this by running simulations where $D=0$ (Figure S8). This assumes there is no environmental diffusion of A so, even if $c>0$, cells will be ‘isolated’ in their own grid location and won’t be able to exchange information with others. We find that in this scenario c does not evolve towards high values because cells don’t gain any benefit from exchanging A with the extracellular space. We now describe this in more detail in lines 265-267:

“... On the one hand, the absence of diffusion prevents the exchange of information between cells and precludes the evolution of high c (which is not beneficial in our model unless cells can communicate with each other. See Figure S8).”

• In the same context: I see that the authors do not intend to consider realistic parameter values with real world units, which is of course ok for a general setting. Nevertheless, one should not completely forget about reality, e.g. by comparing the order of magnitude of c (or the outcome which c would be best for the bacteria in the simulations) with free diffusion – at least c shouldn’t become faster than that. A comment on that would be very helpful.

Indeed, our model is aimed to serve as a conceptual framework to understand how collective sensing could have evolved. In this sense, we are not seeking to mechanistically capture how this process occurred but rather to understand which conditions could promote/impede the benefits of cell-cell communication for environmental sensing. That said, bacteria benefit from collective sensing at values of $c \sim 0.1$, which is lower than the rate of passive transport between neighboring grid cells due to extracellular diffusion - as expected in a realistic scenario. In addition, the optimal value of c in our model is also lower than the upper bound of $c = 3 \text{ min}^{-1}$ estimated for AHLs autoinducers by Kaplan et al (Kaplan & Greenberg, 1985).

• L.220ff.: I think it's difficult to formulate a sentence like that. Maybe there is no benefit of cell-to-cell communication for improving individual estimates of environmental conditions. But that is for sure not the only purpose, as seen in Table 1 and many other publications, there are so many different purposes of the QS mechanism, so it's completely fine if the main benefit for cell-to-cell is more focussed on the local cell density aspect or so.

We fully agree with the reviewer that there are many other benefits of QS beyond collective sensing of the environment. However, this statement is meant to refer to the conditions that are necessary for QS to evolve *only due* to its collective sensing functionality. In the basic version of our model, we intentionally omitted other potential fitness benefits that cells might gain from QS such as collective action or diffusion sensing. As such, we focused on the new functionality we are proposing.

We have now modified the first sentence of the corresponding paragraph to emphasize this. See lines 239-240:

“We find that a series of conditions favor the evolution of QS due to its collective sensing functionality. The first two are related to model assumptions justified previously. ...”

Additional questions: What is meant by "high c"? Compared to what?

In general, for the canonical parameter settings we find that cells are coupled when the mean value of c is larger than 0.1. The specific values are not important but rather that an approximate threshold in c emerges.

• L. 330ff.: I do not fully agree with this general statement, as the whole mechanisms would work even with single cells, completely without exchange, as could be seen at many occasions. This needs to be clarified.

Note that we are not claiming that collective sensing is the only possible function of QS. We fully agree with the reviewer that there are other potential benefits of QS and that for a scenario like diffusion sensing, cells would not have to be together with others to benefit from autoinducer secretion.

With this statement we also want to emphasize that our model shows that collective sensing is a functionality that bacteria can obtain from autoinducer secretion, which *could* have driven QS evolution on its own. As we discuss in the manuscript we believe this could be particularly relevant to systems where bacteria don't use QS to regulate the secretion of extracellular products for which ideas like diffusion sensing or the classical paradigm of cell-cell coordination are much less suitable (see table 1).

• L. 347 ff.: **But this has not much to do with the transporting mechanism (versus free diffusion). It's the general problem the cells may have that they produce molecules which (which mechanism ever) get outside and others react on that. Cheater could still "listen", but save energy i.e. not producing by themselves.**

Thus, it would be essential to take exactly such cheaters into account, in the model setup and then to show that these cheaters have a disadvantage. That would really help for understanding. Here it's "only" shown that QS can develop, but under the (quite restricting) assumption that there are no cheater which use the information of the others without contributing by themselves.

The original version of our model did not allow for the evolution of cheaters. In fact, if a communication cost is implemented the only effect of this parameter would be to prevent the evolution of collective sensing altogether if this cost exceeds the benefit that cells get from correctly estimating the state of the environment (Figure S10). For this reason we have extended our model to decouple the function of A as both the autoinducer and QS product (Figure S11).

Concerning the Supplementary text:

• **A is not a protein (at least not usually for Gram-negative bacteria)**

Thanks for pointing this out. We now refer to it as a molecule, since the architecture of our model is indeed closer to a Gram-negative QS system.

• **I was wondering why the authors let the basal production decrease for large A? Most "classical models" in literature keep it constant and only have the upregulated production (with the positive feedback) dependent on A. This should be adapted or explained, why this modified term is necessary.**

We have simplified the model equations to make the different terms clearer (see equation S1). The basal production rate in the absence of A (i.e. when $A=0$) is constant and given by k_0 . The rate of production of A approaches its maximal value, k , as A increases (with the remaining parameters, K and n , determining the steepness of this transition).

Note that in response to another comment we have reproduced all figures of the manuscript with a set of parameters that is more realistic for QS systems. We now use a much lower Hill coefficient and we assume a larger difference between k_0 and k .

• **A and A_E are mentioned to be concentrations. By that I can understand the terms ($A_E - A$) in the model for the exchange, but to have no conversion factor (for the extracellular versus the intracellular volume) included, obviously means that the same volume is assumed for both, which is not realistic. This needs to be corrected.**

We have now added a term α that equals to $V_{intracellular}/V_{extracellular}$ and thus rescales the concentration of A once it is exported extracellularly. Note that due to the way our model is

constructed, $V_{\text{extracellular}}$ would not correspond to the total volume in the extracellular space but instead to the extracellular volume in the grid space where each cell lives.

If the extracellular volume is much larger than the intracellular volume (i.e. $\alpha \ll 1$) cells won't be able to modify the extracellular concentration of the AI by either secreting or importing AIs. This scenario would impede the evolution of collective sensing because cells are not able to effectively communicate. Nevertheless such small values of α are probably not very relevant to bacterial communities where QS occurs. For instance, similar models (based on experiments in microfluidic chambers where cells are in a two dimensional layer) have yielded a value of α of 1.85 (Dal Co et al 2020).

We now discuss this parameter in the main text and perform additional simulations to understand its role in our model shown in Figure S9. See lines 270-273:

“Likewise, we find that collective sensing is facilitated if bacteria are in moderately confined environments (Figure S9). Otherwise, if the extracellular volume is sufficiently large such that bacterial secretion of autoinducers has little impact on extracellular AI concentration, then cell-cell communication through AI secretion is not effective anymore.”

• Why there is no abiotic degradation of the extracellular A_E included (but for the intracellular A it is)? This needs to be added or explained.

We included a degradation rate for the intracellular concentration of A to guarantee that the system is bistable. Although we don't model this explicitly, this rate can be interpreted –analogous to how it is done in previous models– as the dilution rate of the intracellular components due to cellular growth (this would explain why there is intra but not extracellular degradation). Also, note that autoinducer molecules such as acyl homoserine lactones (AHLs) are actively degraded by several intracellular enzymes so this degradation rate could also result from enzymatic action inside the cell. We now mention this in the section in the Supplementary Information describing the model.

• At least a rough explanation how the parameter values were chosen would be highly appreciated. Does the model behave similarly when other parameter values are applied? E.g. the Hill coefficient seems to be very high with $n=6.75$ (for many typical bacteria it's 2 (having the dimers) or maybe something between 2 or 3). Or k_0 and k only differ by a factor of approx. 5, quite low, often the increased production in bacterial species is observed to be something between 10 and 100.

We chose the original parameters to generate a bistable system with two stable equilibria separated by an unstable equilibrium in the midpoint between both (to not favor any environment over the other) (see Figure S1). We agree with the reviewer that this original choice

is not very representative of QS systems in nature and for this reason we remade all the figures of the manuscript with a lower Hill coefficient and a larger difference between k_0 and k .

• I didn't fully understand: when placing the cells for the next generation: can there cells stay empty, and if yes, how many are that typically? Making this easier to understand / to read would be appreciated.

We agree that this is unclear, especially given the diagram shown in Figure 1. The full grid is repopulated after every cycle so there are no empty locations. We have now clarified this in the figure legend. See lines 179-181:

"...This is illustrated for a single (red outline) high fitness parent and its offspring (fitness increases from green to white). Note that the full grid is repopulated every cycle but for the purpose of illustration only the offspring of one cell is shown . . ."

We have also clarified this in the model description section in the SI. See the third paragraph on page 3.

"At the end of every environmental cycle the fitness of each cell is calculated and the entire grid is repopulated by sampling with replacement $N \times N$ individuals. The probability that an individual is chosen for reproduction equals to its fitness normalized by the total fitness of the population. Upon reproduction, the algorithm for creating and placing the offspring of a cell in the new grid is the following, ..."

Reviewer #2 (Remarks to the Author):

In this paper, the authors develop an evolutionary model to show that quorum sensing can evolve for the function of improving information about the environment when individual information is noisy. Specifically, the authors find that quorum sensing readily evolves when the diffusivity of the extracellular environment is intermediate and environmental heterogeneity is not too small-scaled. I find the model to be concisely tailored to investigate this phenomenon, the paper to be clearly written, and consider the results to be interesting and potentially impactful. I do have some relatively minor concerns, which I outline below. If these concerns are addressed, I think the paper would make a valuable contribution to the literature.

We thank the reviewer for the support of our manuscript and for all the helpful comments. See all our answers below.

1. Explanation of timings or environmental cycling/reproduction in the model (lines 109-21). It is currently not entirely clear what constitutes an 'environmental cycle' in the model and exactly at what point reproduction takes place (and thus with what frequency reproduction occurs relative to the environmental fluctuations). There is both mention of 'non-overlapping generations' (line 126) and of 'bud(ding) multiple times during an

environmental cycle' (line 120). Perhaps this could be explained in somewhat more explicit terms.

We agree that our description could be clearer. Reproduction occurs only at the end of each environmental cycle. We have now clarified this part in lines 125-131:

“...At the end of each environmental cycle the performance of an individual is calculated as the absolute difference between the value of A and the current optimal expression level, either A_{ON} for E_{ON} or A_{OFF} for E_{OFF} , averaged over the cycle duration. This value is then used to compute individual fitness using a sigmoidal function such that cells are penalized for errors in determining whether the environment is in the ON or OFF state, but not for small numerical deviations from the optimal value of A when $c = 0$ (see Figure S2 and Supplementary Information).”

Also see lines 137-144:

“Reproduction occurs at the end of each environmental cycle and cells are selected to reproduce with a probability proportional to their fitness. Reproducing cells are sampled with replacement, and offspring are collected until their total number is sufficient to replace the parental generation and fully repopulate the grid. As a consequence, population size remains constant, generations are non-overlapping and individuals can have multiple descendants in the next generation. Upon cell reproduction, c mutates with probability μ , resulting in c increasing or decreasing with equal probability by a fixed step size δ (subject to the constraint that $c \geq 0$). Finally, daughter cells are placed in the nearest location available to the position of their parent.”

2. Fitness function(line 122 and Supplementary Info). The authors assume a sigmoidal shape of the fitness function so that cells are only strongly penalized for misdetermining whether the environment is in the ON or OFF state but not for small deviations. How did the authors choose the steepness of this function?

The steepness of the fitness function determines the strength of selection for correctly determining the state of the environment: in the extreme case where $s=0$, fitness would be independent of the estimation performance of each cell, whereas for very high values of s , fitness is either 1 or 0 depending on whether a cell estimated the state of the environment correctly or not. Since cells improve their estimation accuracy by communicating through autoinducer secretion, higher values of s will more strongly select for collective sensing to evolve.

Also, note that we chose $s=0.8$. For intermediate values of s -where the fitness curve doesn't have such a pronounced sigmoidal shape (e.g. $s=0.2$)- cells can become penalized for small deviations from the optimal value of A . This acts against small clusters of communicating cells that might make the correct estimate of the environment but take longer to synchronize and thus deviate slightly from the optimal value of A when averaging over the entire environmental cycle.

As a result for intermediate values of s the benefits of communication start appearing for larger clusters of communicators.

We now discuss this parameter in the supplementary materials and included a figure (Figure S5) illustrating how the benefit of collective sensing changes as a function of the steepness of the fitness function. Overall, for less steep functions, collective sensing will take longer to evolve because the fitness benefit resulting from cell-cell communication is lower.

3. Constraints on variable c (lines 127-8 and Supplementary Info). It is mentioned that c is constrained to be equal or larger than 0. Should it not also be constrained to be smaller or equal than 1?

In our model c determines the rate at which the autoinducer diffuses through the membrane by passive diffusion (i.e. in the model equations the internal concentration of A , A_{IN} , changes due to passive diffusion by a rate $c*(A_{OUT}-A_{IN})$). Since c is a rate, its value is constrained to be equal or larger than 0 but it doesn't have any upper bound. We realize this was not well explained in the description of c in the text. We have now clarified this. See lines 108-114:

“In order to study how communication evolves in our model, we let bacteria evolve a parameter c that sets the rate of passive diffusion of A through the cell membrane. This parameter determines the membrane permeability to A and thus the degree of communication between cells (e.g. when $c = 0$, a cell does not share or receive any information from other cells). In nature, membrane permeability depends in part on the biochemical properties of AIs and can change because of variations in AI length or molecular structure, as well as by the evolution of active mechanisms for AI secretion or transport (e.g. carrier proteins) (28–30).”

4. Initial concentration of A (lines 161-2). The authors mention that they draw the initial concentration of A from a uniform distribution of which they specify only the mean. It would be more complete to just give the range of this distribution (based on only the mean we cannot derive this range).

Thanks for pointing this out. We have now specified the interval boundaries. See lines 151-153:

“In particular, we assume that, for each generation of cells, the initial concentration of A is sampled independently for each grid cell from a uniform distribution in the interval $[0, A_{OFF} + A_{ON}]$. ”

5. Cost of permeability (lines 347-8). The authors talk about their assumption that they assume passive diffusion of A across the membrane, but that a cost of permeability (in case of active secretion and sensation) might lead to different results. Do the authors have any ideas on how changes in passive permeability could evolve, as is assumed in their model? If this would occur through changes in the molecular structure of A , this

might then need to be accompanied by mutations in the receptors for A as well. It is easier to see how active secretion and sensation could evolve relatively continuously, but in this case (as the authors note) it would make sense to implement a cost associated with these functions. I think this should perhaps be fleshed out a bit more. Perhaps the authors could explain to what extent their assumptions around this are meant to be realistic, and if they are not meant to be very realistic, how they would expect more realistic assumptions (e.g. involving costs) would change the outcome of the model. Would the system completely collapse because of cheating or would they still expect that there are conditions in which communication could evolve?

We have now explored this issue in more depth in our model by first implementing a cost of communication in the fitness function. Specifically, the fitness of every cell is now multiplied by a term $e^{-c \cdot \text{gamma}}$ where the parameter 'gamma' allows us to vary the strength of the penalty that each cell pays for having a high value of c and thus for communicating with other cells (see Supplementary Information). In the baseline version of the model, collective sensing does not evolve if the cost is higher than the benefit that cells receive for estimating the correct environmental state. However, since we assume that A is also the end product of QS, this model would not allow the emergence of cheater cells that do not produce the autoinducer A but only listen to other cells. To explore the dynamics of cheaters in the context of collective sensing we have extended our original model and used the extended version to explore under which conditions collective sensing can be stable to the evolution of cheaters. See lines 357-378:

“Finally, we asked whether collective sensing would evolve if there is a cost for communication. In the baseline version of our model, where cells must be able to produce AIs to be capable of responding to the environment, a cost of communication will only impede the evolution of collective sensing if it exceeds the benefit that bacteria gain from correctly determining the state of the environment (Figure S10). However, in more complex QS systems, where the autoinducer and the gene product under the control of QS are not the same, it is conceivable that costs of communication may induce the evolution of signal-negative cheaters that do not communicate with the rest of the population but benefit from ‘listening’ to other cells.

To explore the scope for the evolution of such cheaters, we extended our model by incorporating a gene product P whose expression determines fitness and is under the control of A (Figure S11A). In this way, A acts only as the autoinducer and signal-negative mutants can arise without compromising their ability to express alternative phenotypes in the two states of the environment. Simulations of the extended model show that while signal-negative mutants have an advantage in E_{OFF} environments (i.e. when quorum sensing is not needed), they produce insufficient P in E_{ON} environments, where they rely on A from the extracellular environment to upregulate the expression of P (Figure S11B). While signal-negative mutants can compensate for this deficiency by increasing the sensitivity of P expression to A, their spread is eventually halted by negative frequency-dependent selection (Figure S11C). This occurs because signal-negative mutants do not contribute to the extracellular concentration of A, which undermines the effectiveness of collective sensing in their local neighborhood. Accordingly, the rise of signal-negative mutants is also limited when environmental diffusivity is

low, which further contributes to the stability of collective sensing against the evolution of cheating.”

Also we now expanded the discussion on the evolution of cheaters in the context of collective sensing. See lines 412-426:

“One of the main obstacles in explaining the evolution of QS is how this process remains stable in the presence of cheaters. In the context of collective sensing, mutants that do not respond to AIs would not have a fitness benefit because private functions cannot be outsourced to neighboring cells. However, collective sensing could be prone to the evolution of signal-negative mutants that cheat and do not share information but only listen to others. We found that these mutants have a disadvantage in environments that require upregulation of quorum sensing because they rely exclusively on AIs produced by other cells which might not be sufficient for full or timely QS activation. Moreover, even if these cheaters were to compensate for such deficiency by becoming more sensitive to AIs, they are subject to negative-frequency dependent selection and have a limited advantage when there is low diffusivity of AIs in the extracellular space (Figure S11). Since the cost of erroneous decisions regarding the state of the environment is likely higher than the cost of producing AIs, we expect that these different mechanisms will limit the evolution of signal-negative cheaters. Importantly, this further emphasizes that the local nature of bacterial interactions could have favored the evolution of collective sensing and is consistent with previous observations from several QS systems (58, 59).”

Reviewers' Comments:

Reviewer #1:

Remarks to the Author:

Thanks for making many points more clear; also stating more clearly, how the (partially quite restricting) assumptions have been made.

This makes the paper much better understandable.

It's a good contribution to the discussion about "Quorum" sensing and how it could evolve. My doubts are nevertheless a bit in the direction, if the outcome would be still the same with less restricting and very special model assumptions (like the very "narrow" grid structure - of course, one can assume it for a model, but for the big part of bacteria, there will be autoinducer molecules leaving the cells and then diffusion is unavoidable, i.e. there will be always an exchange of information.

From the current explanations, I would more deduce that the main point of the discussion is about private goods versus public goods. (Raising up the question if this couldn't be considered more directly).

Reviewer #2:

Remarks to the Author:

The authors have done an excellent job in addressing my concerns and those of the other reviewer. I especially appreciate the extra work and discussion on cheating. I have no further comments. I congratulate the authors on an excellent article.

Reviewer comments

Reviewer #1 (Remarks to the Author):

Thanks for making many points more clear; also stating more clearly, how the (partially quite restricting) assumptions have been made.

This makes the paper much better understandable.

It's a good contribution to the discussion about "Quorum" sensing and how it could evolve.

My doubts are nevertheless a bit in the direction, if the outcome would be still the same with less restricting and very special model assumptions (like the very "narrow" grid structure - of course, one can assume it for a model, but for the big part of bacteria, there will be autoinducer molecules leaving the cells and then diffusion is unavoidable, i.e. there will be always an exchange of information.

We thank the reviewer for reading our manuscript again and for making more suggestions to improve it. Note that the autoinducer molecules can freely diffuse across grid cells in our model. We only assume that this is not the case to test the effect of extreme extracellular diffusivity values (in this case $D=0$) in Fig S8. We now explicitly refer to the fact that this is a hypothetical scenario in lines 280-282:

"... .However, unlike the monotonic negative effect of cell dispersal, extreme low or high diffusivity hinders the evolution of cell-to-cell communication (Figure 3). On the one hand, in the hypothetical scenario where there is no environmental diffusivity there would be no exchange of information between cells, which in turn would preclude the evolution of high c (Figure S8)...."

Regarding diffusion of autoinducers through the cellular membrane we agree with the reviewer that there might always be a baseline level of diffusion of molecules outside the cell so that the initial value of c is not zero but a slightly higher value. For simplicity we assume that at the beginning $c=0$ but we would expect that a higher initial value for c would just lead to faster evolution of collective sensing

From the current explanations, I would more deduce that the main point of the discussion is about private goods versus public goods. (Raising up the question if this couldn't be considered more directly).

The control of private goods by quorum sensing has previously being explained as a mechanism that stabilizes cooperation in the context of public good production by

preventing the evolution of signal-deaf cheaters. We think that our model offers an alternative and more direct explanation as to why quorum sensing controls private functions: bacteria can make better assessments on whether or not expressing a private trait is beneficial by sensing the environment as a collective through the secretion of autoinducers. We now refer to this distinction in the discussion and provide literature that supports the traditional view of private functions in the context of quorum sensing (see lines 364-369):

“... This functionality can explain why several environmental parameters exert a tight control over the rate of autoinducer production across different quorum sensing systems. Moreover, it offers an alternative explanation as to why bacteria use quorum sensing to regulate the expression of ‘private’ functions which has been previously interpreted as a strategy to stabilize public good-mediated cooperation (9, 52).”

Also we think that our model is best suited to explain the control of private functions by QS (such as competence or sporulation), whereas the classical QS paradigm might be a better framework to understand the control of public goods. We now make this comparison more explicit and refer the reader to papers that perform experimental and theoretical analyses on the role of QS in the control of public goods (lines 369-372):

“Importantly, that bacteria engage in collective sensing does not preclude that cells also benefit from coordinated action (6, 53) and in principle both mechanisms could operate simultaneously in the same quorum sensing system and be relevant for the regulation of private and public functions, respectively.”

Reviewer #2 (Remarks to the Author):

The authors have done an excellent job in addressing my concerns and those of the other reviewer. I especially appreciate the extra work and discussion on cheating. I have no further comments. I congratulate the authors on an excellent article.

We thank the reviewer again for all the helpful comments and for revising the manuscript a second time.